# The Singularity of Cetacea Behavior Parallels the Complete Inactivation of Melatonin Gene Modules

**DOI:** 10.3390/genes10020121

**Published:** 2019-02-06

**Authors:** Mónica Lopes-Marques, Raquel Ruivo, Luís Q. Alves, Nelson Sousa, André M. Machado, L. Filipe C. Castro

**Affiliations:** 1CIIMAR/CIMAR—Interdisciplinary Centre of Marine and Environmental Research, University of Porto, 4450-208 Matosinhos, Portugal; monicaslm@hotmail.com (M.L.-M.); lqalves@outlook.com (L.Q.A.); nelsonfernandosousa@hotmail.com (N.S.); andre.machado@ciimar.up.pt (A.M.M.); 2FCUP—Faculty of Sciences, Department of Biology, University of Porto, 4169-007 Porto, Portugal

**Keywords:** gene loss, Cetacea, melatonin, circadian rhythms

## Abstract

Melatonin, the hormone of darkness, is a peculiar molecule found in most living organisms. Emerging as a potent broad-spectrum antioxidant, melatonin was repurposed into extra roles such as the modulation of circadian and seasonal rhythmicity, affecting numerous aspects of physiology and behaviour, including sleep entrainment and locomotor activity. Interestingly, the pineal gland—the melatonin synthesising organ in vertebrates—was suggested to be absent or rudimentary in some mammalian lineages, including Cetacea. In Cetacea, pineal regression is paralleled by their unique bio-rhythmicity, as illustrated by the unihemispheric sleeping behaviour and long-term vigilance. Here, we examined the genes responsible for melatonin synthesis (*Aanat* and *Asmt*) and signalling (*Mtnr1a* and *Mtnr1b*) in 12 toothed and baleen whale genomes. Based on an ample genomic comparison, we deduce that melatonin-related gene modules are eroded in Cetacea.

## 1. Introduction

Circadian rhythmicity is critical for broad organism homeostasis. In mammals, a complex network of brain anatomical structures, such as the suprachiasmatic nuclei (SCN), the master clock, and the pineal gland, as well as a set of specific molecular pathways, optimise physiological timekeeping [1,2]. First described in the 1950s, melatonin has since been suggested to act as a multi-purpose hormone, with important roles in SCN excitability and sleep modulation [3,4,5,6,7]. In vertebrates, endogenous melatonin levels exhibit daily rhythmicity and a secretory peak at night. Melatonin is predominantly synthesised in the pineal gland, with the canonical synthesis pathway initiating from a precursor molecule, the amino acid tryptophan [3]. The two final steps of the enzymatic cascade are sequentially controlled by the circadian regulated aralkylamine *N*-acetyltransferase (AANAT) and by *N*-acetylserotonin methyltransferase (ASMT) enzymes, respectively, converting the intermediate serotonin into melatonin [8,9]. Subsequent hormone signalling in target cells occurs via two high affinity G-protein couple receptors, (MTNR1A) and (MTNR1B) [10]. While MTNR1A is mostly expressed in the SCN and *pars tuberalis* of the pituitary gland, MTNR1B is predominantly found in the retina, with lower levels detected in the hippocampus and whole brain [10]. Thus, by acting through MTNR1A and MTNR1B receptors, melatonin participates in the light-dependent modulation of SCN activity and sleep/wake cycles [2,11]. Interestingly, vertebrate AANAT functional evolution, towards the conversion of serotonin into the substrate of ASMT, *N*-acetyl serotonin, was suggested to parallel the emergence of the vertebrate pineal gland and melatonin rhythms [12]. In agreement, AANAT is also expressed in the retina, reflecting an ancient role in photoreception [13,14]. Despite the clear presence of a pineal gland in most mammalian species, in some lineages such as mole rats, sirenians and cetaceans, pineal organs were suggested to be absent or vestigial [15,16,17,18]. Consistently, in the naked mole rat (*Heterocephalus glaber*), a subterranean rodent inhabiting a dark ecosystem, an aberrant melatonin system has been described with both melatonin cognate receptors found with inactivating mutations in the coding region [15,19]. Thus, pineal atrophy, such as that described in mole rats, is apparently correlated with genomic gene loss signatures [15,20]. In this context, Cetacea provide a valuable model to track the molecular signature of a macroevolutionary transition, notably regarding the adaptation to a distinct photic and thermal environment [21,22,23]. In fact, the existence of a functional pineal gland in these aquatic mammals is still contentious, even among individuals of the same species: with conflicting reports from various whale species (e.g., *Megaptera novaeangliae* (humpback whale)) or the commonly studied *Tursiops truncatus* (bottlenose dolphin) [16]. To further address the parallelism between pineal regression and melatonin synthesis and signalling, we extracted genomic information and addressed the coding status of *Aanat*, *Asmt* and the melatonin receptors *Mtnr1a* and *Mtnr1b* in Cetacea and in *Hippopotamus amphibius* (common hippopotamus), the closest extant lineage of Cetacea. Further comparative analysis using *Trichechus manatus latirostris* (Florida manatee), also lacking a pineal gland, provided additional insight into the convergent adaptation to fully aquatic lifestyles [18].

## 2. Materials and Methods

### 2.1. Synteny Maps

To perform synteny analyses, the human and *Bos Taurus* (domestic cow) *loci* of *Aanat*, *Mtnr1a*, *Mtnr1b* and *Asmt* genes were used as reference. Importantly, in *B. taurus* two *Aanat* genes were found, and thus we considered as main reference the gene with NM_177509.2 accession number. After collecting the genomic regions of the reference species, several Cetacea genome assemblies and annotations were inspected and scrutinised using the NCBI browser (*Homo sapiens*—GCF_000001405.38, *B. taurus*—GCF_002263795.1, *Orcinus orca* (orca whale)—GCF_000331955.1, *Balaenoptera acutorostrata* (common minke whale)—GCF_000493695.1, *Delphinapterus leucas* (beluga whale)—GCF_002288925.1, *T. truncatus*—GCF_001922835.1, *Lagenorhynchus obliquidens* (Pacific white-sided dolphin)—GCF_003676395.1, *Physeter catodon* (sperm whale)—GCF_002837175.1, *Lipotes vexillifer* (Yangtze river dolphin)—GCF_000442215.1, *Neophocaena asiaeorientalis* (Indo-Pacific finless porpoise)—GCF_003031525.1). To perform the synteny maps the following procedure was used: (1) five flanking genes from each side of the target gene were collected, (2) only genes neighbouring genes classified as coding were considered (Gene Type: Protein Coding), (3) if the target gene was not found, the genomic region between the direct neighbouring genes (first gene at left and right side) was inspected. The inspection was done in two ways: firstly, via blast searches to determine the total or partial absence of the gene and after, manually to ensure the continuity or disruption of the genomic region (with presence or absence of N’s).

### 2.2. Sequence Retrieval

Genomic sequence retrieval for gene annotation was performed using two strategies: (1) In species with fully annotated genomes (*O. orca*, *T. truncatus*, *D. leucas*, *L. vexillifer*, *N. asiaeorientalis*, *P. catodon*, *B. acutorostrata* and *T. manatus latirostris*), the genomic *locus* of the target gene was collected directly from NCBI. The genomic sequence was inspected, from the 5′ neighbouring gene to the 3′neighbouring gene of the target gene. For example, in the case of *Aanat* the genomic sequence was collected ranging from *UBE20* to *RHBDF*. (2) For species whose genomes were not annotated, the corresponding genomic sequence of the target gene was retrieved, from full genome assemblies, via blastn searches using human and *B. taurus* sequences as query.

To scrutinise the status of the *Aanat*, *Mtnr1a*, *Mtnr1b* and *Asmt* gene sequences in *H. amphibius* we built an in-house blast database with the genome assembly (GCA_002995585.1), directly retrieved from the NCBI database. Next, interrogated the in-house database using the human and *B. taurus*, to retrieve full or partial gene sequences for *H.*
*amphibius* (blast-n: -word_size 10, -outfmt 6, -num_threads 50) [24]. The output results were manually scrutinised and, using the qstart, qend, sstart send and bit score options of outfmt6 format of blast software, we identified the scaffolds containing each gene. All scaffolds or sets of scaffolds, with evidence of containing the target genes, were gathered and submitted to AUGUSTUS webserver gene prediction [25]. Finally, the annotation files with the CDS and protein sequence of *Aanat*, *Mtnr1a* and *Mtnr1b* genes were generated (Appendix A). For the *Asmt* gene, only exons 2, 5, 7, 8 and 9 were recovered from the genome assembly. Additional searches in sequencing read archives (SRAs) yielded exon 1 (Appendix A).

### 2.3. Gene Annotation and Mutational Validation

For gene annotation, the collected genomic sequences were uploaded into Geneious7.1.9. (https://www.geneious.com/). The annotation was performed using as reference the corresponding *B. taurus* gene sequences, with the exception of *Asmt* gene, for which the human gene sequence was used as reference due to the presence of multiple expressed isoforms. For each reference sequence, all exons were individualised and mapped to the corresponding genomic region in Cetacea and *T. manatus latirostris*, using the built-in map to the reference tool. The resulting aligned regions were individually inspected to identify ORF abolishing mutations (frameshift, exon deletion, non-canonical splice site and premature stop codons), and concatenated to obtain the predicted sequence. Next, at least one conserved ORF-abolishing mutation was further validated by the identification of the same mutation in different genomic projects or individual samples of the same species. For this, the predicted exonic sequence for each species containing the selected ORF disrupting mutation was used as query in blastn searches in the corresponding available SRAs and Trace Archive in NCBI (only available for *T. truncatus*). Blast hits were collected and uploaded to Geneious and mapped to the corresponding exon. Reads poorly aligned or with identity scores below 95% were removed and the mutational status was confirmed when possible, in at least 2 distinct sequencing projects or individuals from the same species. Validation was performed for all species with the exception of *L. vexillifer* for which only one genomic resource resulting from a single genome sequencing project is presently available in NCBI.

### 2.4. Phylogenetic Analysis

To infer the orthology of *H. amphibius* sequences we performed a phylogenetic analysis using sequences from species belonging to three taxonomic orders: Perissodactyla, Artiodactyla and Primates. Primate sequences were used to root the phylogenetic trees. Briefly, to collect the sequences, blastn searches were done (defaults) in NCBI using as reference the annotated human *MTNR1A * (NM_005958.4), *MTNR1B* (NM_005959.3) and *AANAT* (NM_001166579.1) nucleotide sequences [24]. *H. amphibius* sequences were obtained in the previous analyses (see sequence retrieval section). After sequence collection, an initial alignment was performed and partial and low-quality sequences were removed with Jalview v2.10.5 [26]. A total of 23 *Aanat*, 17 *Mtnr1a* and 18 *Mtnr1b* orthologues were included for further analysis (accession numbers of all datasets can be retrieved from Appendix A). The final orthologous gene sets were aligned independently using MAFFT (setting the parameter—auto mode method selection) [27,28], with the method L-INS-I being automatically selected for all datasets. Using the previous alignments as input, the maximum likelihood phylogenetic analyses were done using the PhyML v3.1 software (http://www.atgc-montpellier.fr/phyml/) [29].To determine the best evolutionary model for each set of sequences, the smart model selection (SMS) option was used [30]: selecting GTR + G + I model for the *Aanat* dataset, and GTR + G model for the *Mtnr1a* and *Mtnr1b* datasets. In addition, to calculate the branch support for each dataset, the aBayes method was selected. Finally, the resulting trees were submitted to the Dendroscope software [31], and the primate clades were used to root the trees.

## 3. Results

### 3.1. *Aanat* Exhibits Conserved Disruptive Mutations in Cetacea

Using the available genomic data we examined the genomic regions of melatonin synthesis and signalling modules in 12 Cetacea species (*O. orca*, *T. truncatus*, *L. obliquidens*, *S. chinensis* (Indo-Pacific humpbacked dolphin), *D. leucas*, *L. vexillifer*, *N. asiaeorientalis*, *P. catodon*, *B. acutorostrata*, *Balaenoptera bonaerensis* (Antarctic minke whale), *Eschrichtius robustus* (grey whale), *Balaena mysticetus* (bowhead whale)) [32,33,34,35,36,37,38,39,40]. We started by analysing the *Aanat* gene. The *Aanat locus-of-origin* was first identified using the human (NM_001088.2) and the *B. taurus* (NM_177509.2) gene sequences (Appendix A). In the case of *T. truncatus*, the *Aanat* gene was not annotated as a coding open reading frame (ORF), yet the gene *locus* and corresponding neighbouring gene families were found to be conserved (Appendix A). Next, we performed a comparative analysis of the exonic sequences, using the cow orthologue as reference, to identify potential mutations [41,42,43,44]. In summary, we were able to deduce the presence of multiple and shared disruptive mutations (Figure 1 and Appendix A). These included a stop codon in exon 1 present in all analysed Cetacea with the exception of *P. catodon* (exon 1 was not found in the current assembly), an absent starting codon and frameshift deletions in exon 3 in Odontoceti (toothed whales), and a disruptive intron 2/exon 3 boundary mutation in Mysticeti (baleen whales). In *P. catodon* a unique deletion of 12 nucleotides in exon 2 was observed, which, combined with an additional 6 nucleotide deletion in exon 3, strongly suggests an inactivated ORF. In *O. orca* and *T. truncatus*, a premature stop codon truncates exon 3. Other species-specific mutations were identified. To further confirm our findings, we selected the most critical and evolutionary conserved ORF-disruptive mutation for validation using SRAs. As such, we confirmed the presence of the premature stop codon in exon 1 across 9 Cetacea species (Appendix A). Regarding the Odontoceti *O. orca*, *T. truncatus*, *D. leucas*, *N. asiaeorientalis* and the Mysticeti *B. acutorostrata* and *E. robustus*, we were able to validate the conserved premature stop using two independent sequenced samples per species.

### 3.2. Cetacea Asmt Reveals Conserved Frameshift Mutation in Exon 1

Next, we examined the *Asmt* gene (Appendix A). In human, 3 distinct isoforms of *Asmt* resulting from the alternative splicing of exon 6 and 7 were identified [45]. Yet, only isoform 1 (P46597) has been reported to display ASMT enzymatic activity (Appendix A) [45]. Thus, both human and *B. taurus* genes were used as reference. Gene annotation revealed the presence of a conserved 1 nucleotide insertion in exon 1, validated by independent SRAs in *O. orca*, *T. truncatus*, *D. leucas*, *N. asiaeorientalis*, *B. acutorostrata* and *E. robustus* (Figure 2, Appendix A). In *L. obliquidens*, *S. chinensis*, *P. catodon* and *B. mysticetus* exon 1 could not be retrieved using the current assemblies. Complete erosion of exon 5 was also detected in *O. orca, L. obliquidens*, *S. chinensis*, *N. asiaeorientalis* and *D. leucas*, for which full genomic sequences ranging from exon 4 to exon 7 were available; for *P. catodon* and *E. robustus* exon 5 was identified with no deleterious mutations, while for the remaining species the presence of sequencing gaps within this region impaired further assessment. The deletion of exon 5 is expected to abolish enzyme activity given that this exon contains a highly conserved glycine (Gly187), participating in the active site of ASMT [45] (Appendix A). Also, aside from the conserved non-canonical splice site observed in exon 8 of Odonoceti, other non-conserved ORF disrupting mutations were found (Figure 2): notably, a 1 nucleotide insertion in exon 4 of *P. catodon*, validated by independent SRAs (Appendix A), and a 1 nucleotide deletion in exon 3 of *L. obliquidens*, also validated by SRA (Appendix A). Finally, exon 9 was not found in the present genome assemblies of all analysed species (Figure 2).

### 3.3. Signalling Gene Modules are Absent or Mutated in Cetacea

We next examined *Mtnr1a* and *Mtnr1b* genes, which are the receptors responsible for transducing melatonin signalling. Here, we did not detect *Mtnr1a*-like sequences in Odontoceti genomes. Comparative synteny analysis revealed that *Fat1* and *F11*, which are adjacent to *Mtnr1a* in human and cow, are also side-by-side in Odontoceti (Figure 3).

To further validate this observation, we carefully examined the genomic sequences flanked by *Fat1* and *F11 genes*. While in some species we observed the occurrence of sequencing gaps, impairing a robust conclusion regarding *Mtnr1a* absence, in *L. obliquidens*, *D. leucas* and *N. asiaeorientalis* the genomic sequences between *Fat1* and *F11* genes are complete, and yield no remnants of *Mtnr1a*. In Mysticeti, we were able to deduce the presence of relics of exon 1, yet presenting potentially disruptive mutations (Figure 4). For instance, in *P. catodon*, a 19-nucleotide deletion in exon 1 was found and validated by SRAs (Appendix A). Regarding exon 2, the current assemblies are incomplete and did not allow a decisive conclusion. Yet, overall, the data strongly suggests that *Mtnr1a* is most likely lost or inactivated across Cetacea.

In the case of *Mtnr1b*, we retrieved *Mtnr1b*-like sequences in all analysed genomes (Figure 5 and Appendix A). Yet, for *L. obliquidens*, only the first exon was identified, while no sequencing gaps were detected between the retrieved exon and the neighbouring *Slc36a4* gene (Appendix A); exon 2 also remained undetected in *O. orca* and *T. truncatus*, yet the presence of sequencing gaps, between the *Mtnr1b*-like sequence and *Slc36a4*, impaired a final conclusion regarding these species. Further examination of retrieved exon 1 and exon 2 sequences indicated a spectrum of mutations with predicted disruptive effects (Figure 5). In exon 1, a conserved stop codon is present in *O. orca* and *S. chinensis*, validated by SRA in the former (Appendix A); a 1 nucleotide insertion in exon 1, generating a premature stop codon, was also found and validated for *B. acutorostrata* and *E. robustus* (Appendix A); and the absence of start codon was detected in *O. orca*, *T. truncatus*, *L. obliquidens* and *S. chinensis*, and further validated by SRA in *L. obliquidens* (Appendix A). In exon 2, a trans-species deletion of 1 nucleotide was found in all species with an intact exon, and confirmed by SRA in all species with the exception of *S. chinensis*, shifting the coding frame from this section of the gene (Appendix A). A 282-nucleotide deletion was also identified in *E. robustus* (Appendix A).

### 3.4. Hippopotamuses Have Coding Melatonin Synthesis and Signalling Orthologues

To further refine our analysis, we next investigated the recently released genome of the *H. amphibius* [39], the closest extant lineage of Cetacea. We were able to deduce the complete ORFs for *Aanat*, *Mtnr1a* and *Mtnr1b* gene orthologues (gene annotations in Appendix A). In contrast, in the case of *Asmt* we were able to retrieve partial sequences (exons 2, 5, 7, 8 and 9). Furthermore, searches in SRAs allowed the recovery of exon 1 (Appendix A). Regarding *Aanat*, two *Aanat*-like genes were retrieved for *H. amphibius* as well as *B. taurus* and *Capra aegagrus hircus* (domestic goat), but these are lineage-specific duplications (Appendix A). Phylogenetic analysis of *Aanat*, *Mtnr1a* and *Mtnr1b* gene sequences revealed that *H. amphibius* orthologues clustered within Artiodactyls (even-toed ungulates), as expected (Appendix A). These findings indicate that the inactivation of melatonin-related pathways most likely took place in the common ancestor of Cetacea, after the divergence of Hippopotamidae and Cetacea.

### 3.5. Convergent Disruption of Melatonin Metabolism and Signalling Modules in the Fully Aquatic Manatee

We next examined an additional exclusive aquatic non-cetacean mammal, *T. manatus latirostris* and identified ORF-abolishing mutations in all melatonin synthesis and signalling module genes (Appendix A). Regarding *Aanat*, these included an absent start codon and a canonical splice site mutation in exon 3, whereas in *Asmt*, a premature stop codon was identified in exon 5. In *Mtnr1a*, a single nucleotide deletion was identified at the exon 2 and a set of frameshift mutations including a premature stop codon were identified in the exon 2 of *Mtnr1b*. The identified mutations were further validated by SRAs (Appendix A).

## 4. Discussion

Taken together, our results strongly suggest that melatonin production and signalling is impaired in Cetacea (Figure 6). Despite the lower number of disrupting mutations found in *Asmt*, we still find indications of pseudogenisation in the analysed species with the exception of *B. mysticetus*, due to poor genome coverage. Furthermore, we were able to infer that ancestral inactivating mutations (shared between toothed and toothless whales), occurred in *Mtnr1b* and *Aanat* ORFs. In the absence of compensatory mechanisms, the observed pattern of gene loss, or coelimination of functionally related genes could represent a case of regressive evolution [46]. Convergent disruption of this set of genes was also observed in *T. manatus latirostris*. These evidence support the anatomical observations that the pineal gland is absent or vestigial and that classical melatonin-based circadian rhythmicity is lacking in these fully aquatic species [16,17].

Yet, a paradoxical observation emerging from our findings relates to the reported presence of circulating melatonin in the bottlenose dolphin [16]. Without a functional AANAT or specific receptors, it seems unlikely that the reported melatonin levels in the bottlenose dolphin are endogenously and rhythmically produced by the canonical cascade and/or act on MTNR-dependent pathways. Thus, melatonin levels could be explained by alternative metabolic pathways: i.e., an increased activity of other enzyme cascades, including AANAT-independent pathways. In effect, AANAT-independent melatonin synthesis has been previously reported, notably in peripheral organs [47,48]. For instance, in mouse skin samples, serotonin acetylation into *N*-acetylserotonin was shown to be mediated by the more promiscuous arylamine *N*-acetyltransferase (NAT), in the absence of AANAT [47,48]. Moreover, an alternative cascade, with reversed enzymatic steps, has also been proposed. In this scenario, serotonin is initially methylated, yielding 5-methyltryptamide, which can serve as substrate for AANAT or NAT to be further *N*-acetylated into melatonin [47]. Given the circadian regulation of AANAT activity, and the lower stability of 5-methyltryptamide, when compared to *N*-acetylserotonin, the classical pathway is suggested to prevail regarding pineal melatonin build-up [13,14,47]. An ubiquitous ASMT-like gene has also been identified in vertebrates, with an unreported function so far [49,50]. The remarkable conservation of the putative catalytic domain for SAM binding supports the methyltransferase activity of this ASMT-like enzyme [50]. On the other hand, melatonin could be acquired from food sources [4]. In fact, no circadian melatonin fluctuations were indisputably acknowledged for bottlenose dolphin [16].

Regardless of the putative maintenance of endogenous, pineal-independent, melatonin sources, our results strongly support the inactivation of both MTNR1A and MTNR1B receptors in Cetacea, and in the Sirenia *T. manatus latirostris*. What would be the trade-off of losing melatonin synthesis and signalling modules? Melatonin is commonly known for participating in the maintenance of circadian rhythms. Yet, in animals with impaired melatonin synthesis, sleep/wake cycles are not fully disrupted [51]. Nonetheless, melatonin does seem to have synchronising and phase-locking effects, allowing an optimised adjustment to light/dark cycles [51]. In fact, deletion of both *Mtnr1a* and *Mtnr1a* in mice suggested a cooperative role in the regulation of sleep/wake cycles, notably through the increase in wakefulness [11]. In addition, *Aanat* knock-out mice were also shown to exhibit an increased diurnal activity [51]. Thus, the erosion of the melatonin synthesis and signalling modules could have accompanied cetacean idiosyncratic sleeping and vigilant behaviours [52,53]. This loss might also reflect other specific life-history traits. For instance, fully aquatic mammals dwell in light-limited, and wavelength-selective, environments with specific thermal constraints [23]. Both light intensity and spectral properties were suggested to modulate melatonin production [22,54]. In addition, melatonin-induced sleep entrainment promotes body temperature shifts [21]; thus, the offsetting of melatonin production and signalling in aquatic mammals could facilitate thermoregulatory mechanisms.

In the absence of melatonin receptors, circulating melatonin could act on MTNR-independent physiological processes, such as inflammatory and immune responses suggested to be modulated by melatonin-dependent gene-expression [4]. Also, one cannot disregard the ancient antioxidant capacities of melatonin [4]. In fact, cetaceans have evolved strategies to offset hypoxia-induced oxidative stress, mirrored by the mutational tinkering observed in mink whale’s genes participating in antioxidant systems (e.g., glutathione metabolism [40]). Overall, our findings provide a unique genomic signature suggesting a behavioural adaptation entailed by a cascade of gene loss events in Cetacea [55].

## Figures and Tables

**Figure 1 genes-10-00121-f001:**
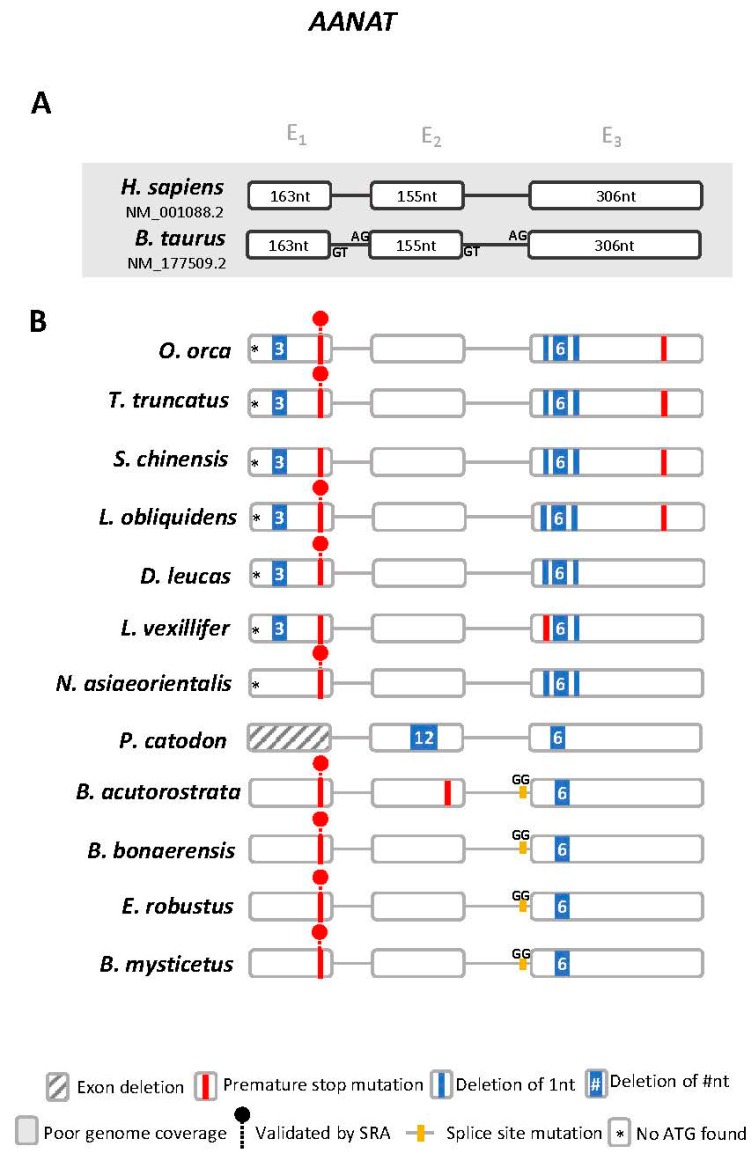
*Aanat* gene annotation. (**A**) Schematic representation of the gene structure of human and *Bos taurus Aanat* gene, each box represents an exon and lines represent intronic regions (not to scale). (**B**) Schematic representation of the corresponding *Aanat* genes identified in Cetacea and location of the identified mutations. Non-canonical splice sites are indicated above the corresponding annotation. SRA: sequencing read archives.

**Figure 2 genes-10-00121-f002:**
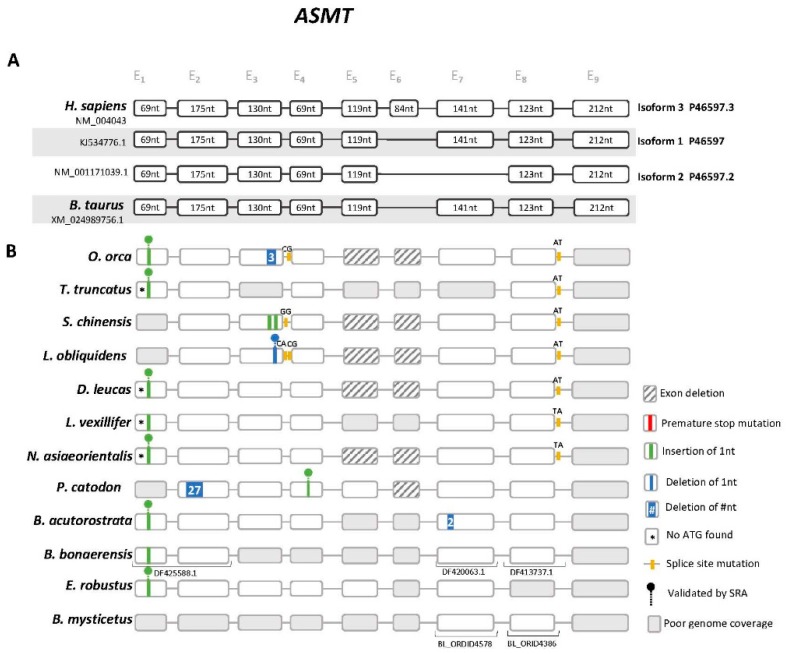
*Asmt* gene annotation. (**A**) Schematic representation of the gene structure and distinct isoforms of human and *B. taurus Asmt* gene (not to scale), each box represents an exon and lines represent intronic regions. (**B**) Schematic representation of the corresponding *Asmt* genes in Cetacea and location of the identified mutations. Non-canonical splice sites are indicated above the corresponding annotation.

**Figure 3 genes-10-00121-f003:**
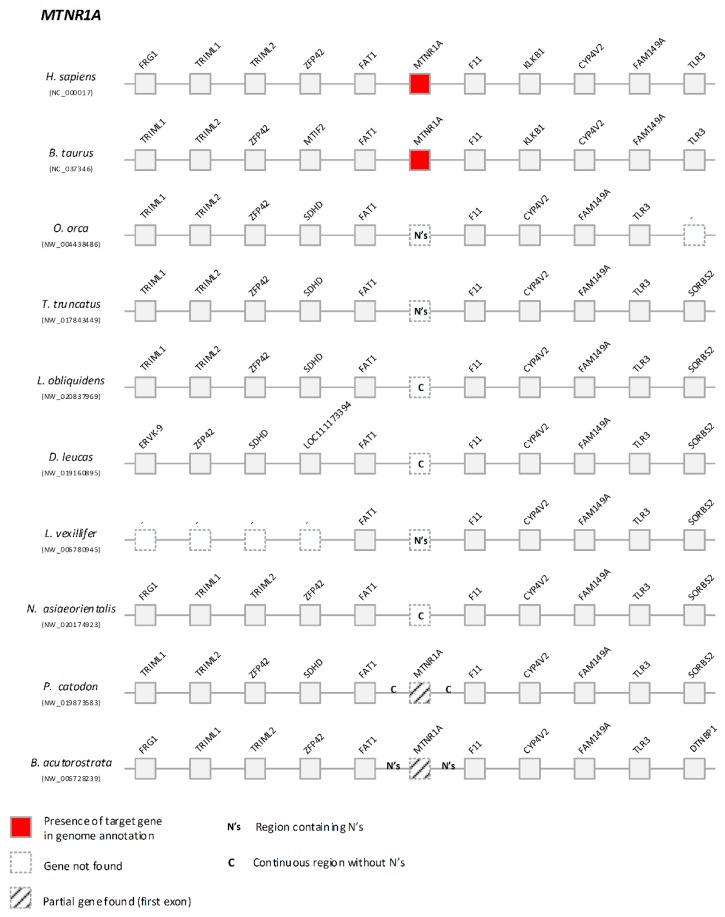
Comparative synteny maps of *Mtnr1a* genomic locus.

**Figure 4 genes-10-00121-f004:**
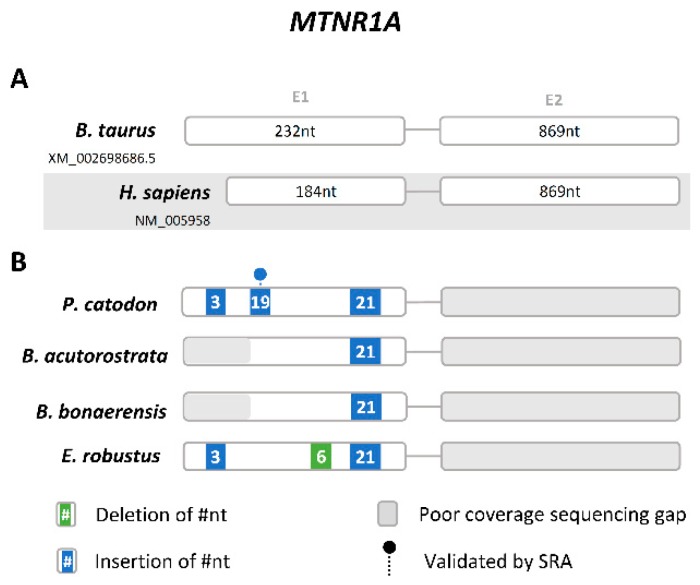
Gene annotation of *Mtnr1a* relics in Mysticeti. (**A**) Schematic representation of the gene structure of human and *B. taurus Mtnr1a* gene (not to scale), each box represents an exon and lines represent intronic region. (**B**) Schematic representation of the corresponding *Mtnr1a* genes identified in Cetacea and location of the identified insertions and deletions.

**Figure 5 genes-10-00121-f005:**
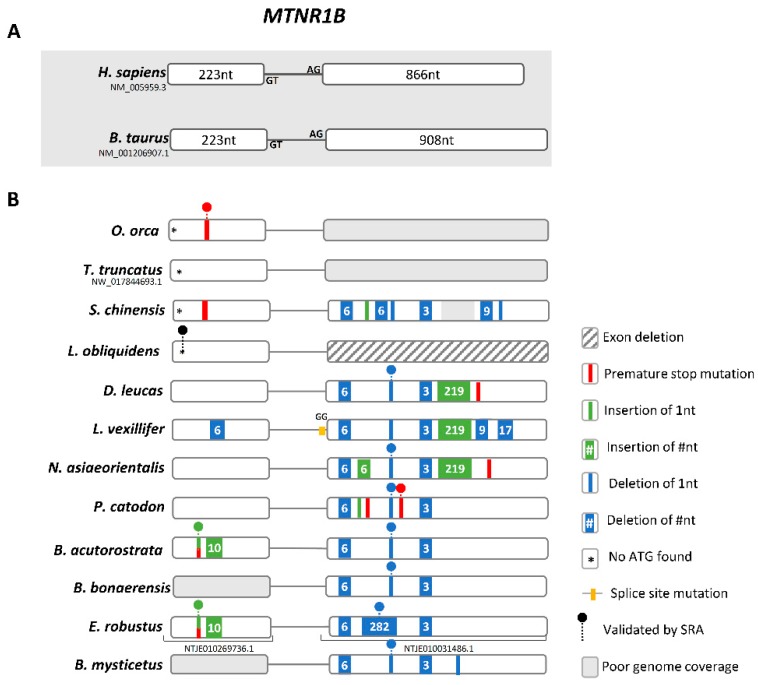
*Mtnr1b* gene annotation. (**A**) Schematic representation of the gene structure of human and *B. taurus Mtnr1b* gene (not to scale), each box represents an exon and lines represent intronic region. (**B**) Schematic representation of the corresponding *Mtnr1b* genes identified in Cetacea and location of the identified mutations. Non-canonical splice sites are indicated above the corresponding annotation.

**Figure 6 genes-10-00121-f006:**
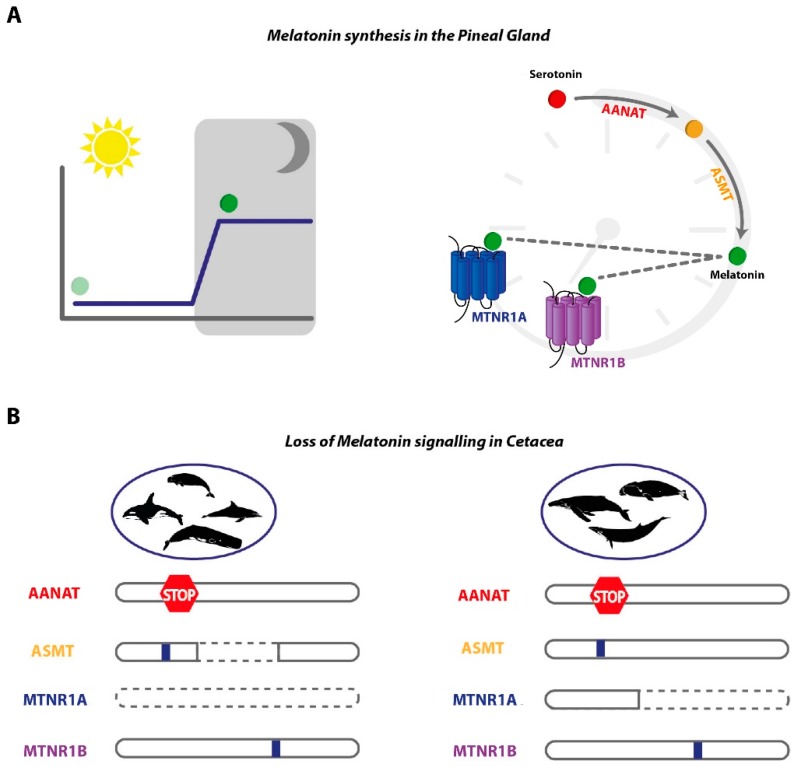
Melatonin synthesis and signalling modules are eroded in Cetacea. (**A**) Melatonin synthesis exhibits daily rhythmicity with a production peak at night (left); AANAT and ASMT sequentially convert serotonin into melatonin, which serves as ligand to two high affinity G-protein couple receptors, MTNR1A and MTNR1B (right). (**B**) Analysed genes from toothed whales (Odontoceti, left) and baleen whales (Mysticeti, right), premature stops, alternative splicing, indels (blue box), and complete gene erosion (dashed line), are depicted.

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
