# Peer review of "The Singularity of Cetacea Behavior Parallels the Complete Inactivation of Melatonin Gene Modules"

_genes, 2019, doi:10.3390/genes10020121_

Round 1
Reviewer 1 Report
In the current study, the authors report that the singularity of Cetacea behavior parallels the complete inactivation of melatonin gene modules. This is a very interesting study and provides novel information on melatonin research. I have some minor comments. The data strongly suggest that the melatonin membrane receptors MT1 and MT2 are absent or completely inactive, particularly the MT1. This indicates that melatonin may not participate in the regulation of biorhythm in cetacea, at least, not being mediated by melatonin receptors. The genes of MT1 and MT2 are probably lost during evolution for some reasons. This should be discussed in more details.
As to the melatonin production in cetacea it seems not as simple as the authors expected. The data in the current study indicated that AANAT may not be functional at all due to the premature stop mutation, but the ASMT has the high chance to be functional with some insertion and deletion. It has been reported an alternative pathway for melatonin synthesis in which, the ASMT can convert serotonin to 5-methoxytryptamine. Then many acetyltransferases can convert 5-methoxytryptamine to melatonin, for example NAT. The NAT has been identified in human skin cells. Thus, the cetacea still have the capacity to produce melatonin. This can be easily proved by testing the melatonin production in cultured whale cells (any cells) since melatonin can be synthesized in all cells with mitochondria. This should also be discussed in details.
Line 33-34: “hydroxyindole-O-methyltransferase (ASMT)” should be “(HIOMT)” or change to “N-acetylserotonin methyltransferase (ASMT)”.
Author Response
We thank the reviewer for his/her comments and suggestions. In the present version of the manuscript we provide additional introductory background, notably on melatonin receptor function, as well as a more detailed discussion on the possible outcomes of receptor signalling loss. To that aim, we included further information, such as previously published data obtained from receptor knock-out models. Also, we further discuss the environmental cues that possibly underscore melatonin-related gene inactivation. Regarding the alternative synthesis pathways, we expanded the discussion to provide more detail on the non-canonical cascades. Yet, we do emphasize that circadian melatonin build-up is most likely liked to the classical AANAT/ASMT pathway. Moreover, given the clear inactivation of melatonin receptors, circulating melatonin levels are unlikely to participate in sleep/wake cycles. In agreement, alternative roles attributed to melatonin are listed. The ASMT full name was corrected.
Reviewer 2 Report
Review of “The singularity of Cetacea behavior parallels the complete inactivation of melatonin gene modules” by Mónica Lopes-Marques, Raquel Ruivo, Luís Q. Alves, Nelson Sousa, André M. Machado and L. Filipe C. Castro.
Lopes-Marques et al. have analyzed four genes (Aanat, Asmt, Mtnr1a and Mtnr1b) related with melatonin synthesis and signaling in 12 cetacean genomes. They have convincingly demonstrated that these genes have been eroded in the analyzed species, which might be related with pineal regression and a distinctive bio-rhythmicity.
I think that the manuscript is clear and well written, the methodology is adequate, the analyses are exhaustive and convincing, and the discussion of the results and resulting conclusions are appropriate. I think that the work of Lopes-Marques et al. is attractive for scientists interested in the evolution of the melatonin signaling in mammals, and also for scientists interested in the analysis of gene loss events and their impact on the evolution of species. I think that the manuscript is therefore suitable to be published in Gene.
I only have one major point and two minor suggestions:
My major concern is about the phylogenetic analyses. These analyses are usually useful for supporting the orthology of the analyzed genes. In this work, however, the authors support the orthology based on the synteny maps, which is a convincing data, but they do not include the cetacean sequences in the phylogenetic reconstructions. This implies that the phylogenetic results are irrelevant for the aim of the work. In addition, the resolution of the phylogenetic trees is poor, and some groups are unexpected. Just to mention one example, the Mtnr1a sequence of the bat Desmodus rotundus clusters closer to primates Mtnr1a sequences than to other Chiroptera Mtnr1a genes (overall, it is expected that gene trees replicate the species trees). Because the aim of the work is not to solve the phylogeny of the Aanat, Asmt, Mtnr1a and Mtnr1b genes in mammals, I would suggest: 1. to include the cetacean sequences in the analyses, and 2. to reduce the number of the mammalian sequences, focusing in the species relevant for the work (e.g. cetaceans, hippopotamus and artiodactyls, and some other mammalian species (maybe primates) as outgroup). These analyses might provide valuable information for the work since it would expect long branches for the cetacean sequences grouping with the hippopotamus genes (or even outside due to a long-branch attraction artifact), which would support the idea that cetacean sequences are non functional genes evolving faster that the functional ones. This result might be included in the Results and in the Discussion sections of the manuscript.
I the case that it is not possible to obtain the phylogenetic reconstructions with the cetacean sequences, and the phylogenetic trees cannot be improved resolving the conflicting groupings, I suggest to remove them from the work.
Minor suggestions/points:
1. In the Introduction, authors mention that other mammalian species lack a structured pineal gland. I think that these species includes manatees and some toothless animals, for which genomes projects are available. To provide a wider impact of the results (see for instance Sharma et al. Nature Communications doi: 10.1038/s41467-018-03667-1), it would be interesting to know the status of Aanat, Asmt, Mtnr1a and Mtnr1b genes in these lineages. Could data from these species be included in the manuscript? In the case that these species also have lost these genes, the authors might argue about ‘convergent gene losses’. I also think that the discussion of the results might be extended, including concepts such as ‘coelimination’ of functionally related genes or scenarios of ‘regressive evolution’ (see for instance Marti-Solans et al. Mol Biol Evol doi:10.1093/molbev/msw118).
2. The numbers of the References are wrong. I guess that the numbers come from a previous version of the manuscript in which the Material and Method section was the last one. Please, correct them.
Author Response
We thank the reviewer for his/her comments and suggestions. We agree that the resolution of the provided trees was poor. The central aim of our phylogenetic analysis was to assign the orthology of the retrieved H. amphibius sequences. In the present version of the manuscript we have followed the reviewer’s suggestion and limited the mammalian sequences to Perissodactyla, Artiodactyla and Primates, excluding the annotated cetacean pseudogenes (given their eroded nature). Thus, the materials and methods were revised accordingly and the new trees and tables can be found in the revised version of the supplementary information file. We have also included the fully-aquatic Florida manatee in our analysis and observed a convergent disruption of melatonin metabolism and signalling genes. We thank the reviewer for this suggestion. These novel results are included and discussed in the present version. Figures can be found in the revised supplementary information file. We have also included the suggested concepts of “coelimination” and “regressive evolution” in the revised Discussion. The references were verified and corrected.